# OpenReview forum: "Stabilizing Reinforcement Learning for Honesty Alignment in Language Models on Deductive Reasoning"
_ICLR.cc/2026/Conference — Submitted to ICLR 2026_

### Official Review · Reviewer_YE4W · 2025-10-31

**Soundness:** 2
**Presentation:** 2
**Contribution:** 2
**Rating:** 2
**Confidence:** 4

**Summary:**

The paper addresses the problem of abstaining giving an answer in deductive reasoning when the premises arent sufficient to answer a question. The authors show that untrained LLMs (albeit small LMs) are unable to do the task well as task complexity increases. The authors introduce ANCHOR; that adds a complementary SFT objective to GRPO to train models that can abstain.

**Strengths:**

- The motivation is clearly presented and the problem is relevant: Models should know when a question is unanswerable and rather than producing a confident incorrect answer
- I like the set of tasks used in the paper to test the methods. Being able to systematically vary difficulty is a neat setup. Correspondingly, I like figs 1,2
- The paper does a good job of evaluating existing models on their proposed tasks.

**Weaknesses:**

- While the synthetic tasks provide a good knob to tune difficulty / complexity, I dont know how realistic the task setup is. The queries seem overly complicated, unnatural, and distant from how llms are used.  The unanswerable cases are created through edge removal etc. that may not reflect how unanswerable questions arise. Adding a more realistic / conventional task might help the paper with ecological validity.
- My main concern are the counterintuitive GRPO+SFT results. Why does RL actively make the SFT’ed model worse? I also have concerns around the implementation of GRPO:
    - Small group size of 5
    - Very Off-policy: (global batch 1024, mini-batch 64, micro-batch 2)
- The paper views honesty pretty narrowly as abstention.
- ANCHOR requires access to ground truth trajectories during the RL phase which might be an unrealistic requirement.

**Minor:**
- place related work in the main draft
- add examples of the task in the main, to improve clarity

**Questions:**

- How do stronger models perform on these tasks? Either open source frontier models or GPT-5 / Claude
- How is the data for sft generated?
- Could SFT+GRPO with a stronger KL penalty (e.g., 0.01 instead of 0.001) recover ANCHOR-like behavior? Since ANCHOR forces the model to be close to the SFT data.

---

> ### Author Response · Authors · 2025-11-18
>
> Thank you for your feedback. In addition to our general response, we address your specific concerns below.
>
> > 1. Realism of the synthetic tasks and ecological validity.
>
> Thank you for raising this concern. As elaborated in our general response, our primary goal is to study honest deductive reasoning in a way that isolates reasoning from factual recall. This requires task formulations where external world knowledge cannot shortcut the reasoning process. For this reason, we intentionally design the datasets so that the model must rely on explicit multi-step reasoning provided in the prompt.
>
> As an example, although the underlying GraphLA tasks correspond to systems of linear equations, we express each equation through dish-price comparisons in natural language. This prevents models from solving the entire system using symbolic shortcuts (e.g., Cramer’s rule) and ensures that each step must be deduced explicitly. Across all evaluated models, we did not observe any behavior indicating the use of latent algebraic knowledge, which gives us confidence that the tasks do isolate deductive reasoning.
>
> Regarding unanswerable instances: edge removal is one meaningful way to break a reasoning chain by removing a critical clue, analogous to missing premises in real-world multi-step reasoning. In addition to edge removal, we also apply several dataset-specific perturbations to generate varied unanswerable cases. We welcome suggestions for additional perturbation methods that may be useful in this setting.
>
> > 2. Counterintuitive GRPO+SFT results and concerns about GRPO implementation.
>
> Conceptually, this counterintuitive arises because GRPO optimizes a fundamentally different objective than SFT, so starting RL from an SFT checkpoint introduces a distribution shift. When GRPO’s reward structure conflicts with the SFT-induced behavior, especially under difficult datasets where incorrect rollouts dominate. It can push the model away from the SFT solution.
>
> Regarding implementation details: the batch sizes and parallelism configuration reflect the limits of our available compute, and they were selected based on pilot experiments exploring the efficiency–performance trade-off. Similar hyperparameter regimes are also used in existing work [1-3]. More broadly, modern “on-policy RL” methods for LLMs are effectively off-policy to some degree unless all rollouts in the global batch are generated by the current policy, which is not feasible under typical training setups.
>
> > 3. The paper views honesty narrowly as abstention.
>
> Thank you for pointing this out. Our use of the term honesty is specific to the setting we study: determining whether a deductive question is answerable based solely on the given premises, and abstaining when it is not. While abstention can also occur in safety contexts, we treat those cases as belonging to the “harmlessness” domain rather than honesty. Among the available terminology, “honesty” most accurately captures the behavior we aim to measure: acknowledging what the model does and does not know based on provided information. We clarify this distinction further in Appendix A.
>
> We have also expanded and reorganized the related work section in the main text to discuss relevant prior work more comprehensively, and we added more examples to improve clarity. We hope these clarifications address your concerns, and we appreciate your review. We would be grateful for your consideration in updating the score.
>
>
> [1] https://arxiv.org/abs/2402.03300
> [2] https://arxiv.org/abs/2503.09516
> [3] https://arxiv.org/abs/2506.05316

---

> > ### Author Response · Authors · 2025-11-20
> >
> > Dear reviewer, we would like to kindly ask if we have addressed your concerns. We sincerely appreciate your questions and feedback!

---

> > > ### Author Response · Authors · 2025-11-21
> > >
> > > Dear reviewer, we sincerely appreciate the time you have invested in reviewing our work. If you have any further thoughts after reading our rebuttal, we would be grateful to hear them.

---

> > > > ### Author Response · Authors · 2025-11-22
> > > >
> > > > Dear reviewer, we appreciate that you may have limited time to review our responses. Please feel free to share any thoughts or suggestions at your convenience.

---

> > > > > ### Author Response · Authors · 2025-11-25
> > > > >
> > > > > Dear reviewer, we hope you are doing well. We wanted to kindly check in to see if there is anything further we can clarify regarding our work. If you find our responses helpful, we would be grateful if you would consider updating your score. Thank you again for the time and care you have dedicated to reviewing our submission.

---

> > > > > > ### Author Response · Authors · 2025-11-27
> > > > > >
> > > > > > Dear reviewer, the author-reviewer discussion phase will end soon. We warmly invite you to consider our responses and raise your scores if our responses were helpful. We are very happy to provide further information and thank you very much for your time!

---

### Official Review · Reviewer_n28i · 2025-10-31

**Soundness:** 2
**Presentation:** 2
**Contribution:** 3
**Rating:** 4
**Confidence:** 4

**Summary:**

This paper addresses the challenge of honesty alignment in LLMs—that is, ensuring models accurately indicate what they do and do not know. To overcome the scarcity of relevant data, the authors developed two new datasets. They evaluated three Qwen models on these benchmarks and observed that performance deteriorates as task complexity increases. Finally, the authors introduced *ANCHOR*, an approach that incorporates ground-truth trajectories into reinforcement learning algorithms.

**Strengths:**

1. The paper is well-written and easy to follow.
2. The study on the reasoning tasks of variying deductive difficulty is well-designed and provides good insights.
3. The proposed datasets for honesty alignment are a good contribution to the community.

**Weaknesses:**

1. The statement that "when GRPO has all incorrect rollouts for one question, the so-called 'gradient vanish' problem will prevent any learning progress" is somewhat debatable:
    - The zero-advantage issue holds for a single question. However, in actual optimization, a mini-batch usually covers several questions, so having all zero advantages in a mini-batch is rare, and optimization can still proceed.
    - The Adam/AdamW optimizer retains momentum, which allows optimization to continue even if a zero-gradient occurs rarely in a mini-batch.
    - If the curriculum learning hypothesis—discussed in the paper—holds, GRPO will not completely prevent learning progress unless all questions are unanswerable. However, this may come at the cost of reduced training efficiency. The result of GRPO+Easy-to-Hard can provide some insight here.
    - This issue pertains only to GRPO, not to all RL post-training techniques (e.g., PPO, ReMax). Discussion of this distinction is necessary.

2. When forcing the ground-truth trajectory, the advantage of RL over SFT—namely, going beyond the dataset distribution—is likely diminished, as the LLM is being compelled to follow specific reasoning paths.
   - There is a lack of comparison with SFT + Ground-Truth trajectories. SFT might be more efficient while sharing the same limitation.
   - For larger LLMs (i.e., Qwen-3B), GRPO + Easy-to-Hard already achieves better performance, which may hint at the issue raised here.


3.  The paper only conducts experiments on the small LLMs (0.6B, 1.7B, 3B). Considering the issue mentioned in W2, the results may not generalize to larger models (7B, 32B, 70B).

4. In the evaluation, the authors claim that the tasks are deductive; i.e., the prompts are self-contained and provide all necessary information to answer the questions. However, this ignores the internal knowledge of the LLMs. What if the LLMs rely on their own knowledge to solve the task? How can we determine the behavior of the LLMs?

5. The paper presents a variety of design choices, but each is not discussed in sufficient depth. In particular, the curriculum learning technique (Easy-to-Hard) appears to address most of the issues raised in the motivation—provided the base model is not too weak relative to the tasks. A more thorough discussion of the necessity and contributions of each individual design would strengthen the paper.

> I personally think this work makes a valuable contribution and has notable strengths, although it also retains some unignorable weaknesses.

**Questions:**

1. Need to clarify the LLM usage, as it is used as the target in research.
2. Wrong results on Table 1? GRPO gets 1 on the unanswerable set and 0 on the answerable set.

---

> ### Author Response · Authors · 2025-11-18
>
> Thank you for your thoughtful feedback. In addition to our general response, we address your specific concerns below.
>
> > 1. The statement that “when GRPO has all incorrect rollouts for one question, the so-called ‘gradient vanish’ problem will prevent any learning progress” is somewhat debatable.
>
> We understand the concern that obtaining all zero advantages within a mini batch may be rare in other settings. However, note that the datasets used in RQ2 are substantially more challenging than those in RQ1. As a result, the initial model performance is extremely low. For example, on GraphLA, the initial accuracy on the answerable subset is below 0.01. In practice, we observed that this level of performance is insufficient to generate gradients that guide the learning process toward correct reasoning. Instead, the model consistently exploits the reward by answering “Unknown” for all examples, without attempting to follow the intended reasoning steps. Under these conditions, neither Adam nor other optimization techniques enabled GRPO to escape this failure mode. Our empirical results therefore reflect a dataset-induced regime where GRPO cannot initiate meaningful learning progress.
>
> > 2. When forcing the ground truth trajectory, the model may be compelled to follow specific reasoning paths. There is also a lack of comparison with SFT + ground truth trajectories.
>
> We appreciate this point. In ANCHOR, however, the ground truth trajectory serves as dynamic guidance rather than a fixed target. At intermediate stages of training, when the model is able to produce correct rollouts on its own, both the on-policy and ground truth rollouts receive positive rewards. In such cases, ANCHOR does not force the model toward the off-policy trajectory, unlike SFT. The benefit is most pronounced during early stages on difficult datasets, which is our primary focus.
>
> Regarding “SFT + ground truth trajectories,” we may not fully understand the configuration you have in mind, since SFT already uses the ground truth trajectory as the output label. If you had a different variant in mind, we would be glad to learn more.
>
> > 3. Experiments are limited to smaller LLMs (0.6B, 1.7B, 3B), and results may not generalize to larger models.
>
> In our experiments, we observe that larger models indeed perform better on harder dataset variants. Since our deductive reasoning formulation can scale to arbitrarily many steps, there is no inherent difficulty ceiling. For any model size, one can construct a dataset version for which the model’s initial performance is nontrivial. For RQ1, we also evaluated Qwen 3 8B and found that reasoning depth $𝑘=20$ already leads to complete failure under prompting. Even if a fixed dataset happens to be solvable for some larger models via SFT, this is orthogonal to our goals. It would be more informative to study how each method performs as dataset difficulty grows alongside model capacity. In our current paper, the selected dataset parameters already result in sufficiently challenging tasks for all three model sizes, so further increases would impose unnecessary sequence length and compute costs without changing the underlying conclusions.
>
> > 4. The deductive reasoning tasks ignore internal knowledge of LLMs. How do we know models do not rely on latent knowledge?
>
> This is an important question and directly motivates our dataset design. Our goal is to study honest deductive reasoning isolated from factual recall. To this end, we deliberately construct prompts that require stepwise reasoning rather than allowing direct formulaic shortcuts. For example, although the system of linear equations underlying GraphLA could be solved at once using Cramer’s rule in symbolic form, the natural language formulation using dish-price comparisons eliminates this possibility. In all evaluated models, we did not observe generations that bypassed the intended reasoning process. This gives us confidence that the tasks isolate deductive reasoning rather than tapping into pretrained factual knowledge.
>
> > 5. The curriculum learning technique (Easy-to-Hard) appears to address most of the issues raised, assuming the base model is not too weak.
>
> As noted in item 3, dataset difficulty is not fixed. This applies equally to curriculum learning. Designing the “easy” version is nontrivial. If the easy set is too easy, its distribution may be too far from the target dataset to provide useful transfer. If it is too hard, it effectively becomes a hard dataset itself, and GRPO may still fail. Our discussion in Lines 458 to 466 elaborates on these trade-offs. We would be happy to provide additional details if useful.

---

> > ### Author Response · Authors · 2025-11-18
> >
> > > 6. Potential error in Table 1: GRPO gets 1 on the unanswerable set and 0 on the answerable set.
> >
> > The reported numbers are actually correct. Unanswerable instances are trivial for the model, which can reliably predict “Unknown” without performing any reasoning. This causes GRPO to exploit the reward structure and degenerate into always predicting “Unknown,” leading to perfect performance on the unanswerable subset but complete failure on the answerable subset, even though the dataset is balanced.
> >
> > We hope these clarifications address your concerns, and we appreciate your review. We would be grateful for your consideration in updating the score.

---

> > > ### Author Response · Authors · 2025-11-20
> > >
> > > Dear reviewer, we would like to kindly ask if we have addressed your concerns. We sincerely appreciate your questions and feedback!

---

> > > > ### Author Response · Authors · 2025-11-21
> > > >
> > > > Dear reviewer, we sincerely appreciate the time you have invested in reviewing our work. If you have any further thoughts after reading our rebuttal, we would be grateful to hear them.

---

> > > > > ### Author Response · Authors · 2025-11-22
> > > > >
> > > > > Dear reviewer, we appreciate that you may have limited time to review our responses. Please feel free to share any thoughts or suggestions at your convenience.

---

> > > > > > ### Author Response · Authors · 2025-11-25
> > > > > >
> > > > > > Dear reviewer, we hope you are doing well. We wanted to kindly check in to see if there is anything further we can clarify regarding our work. If you find our responses helpful, we would be grateful if you would consider updating your score. Thank you again for the time and care you have dedicated to reviewing our submission.

---

> > > > > > > ### Comment · Reviewer_n28i · 2025-11-25
> > > > > > >
> > > > > > > This version corrects the grammar while maintaining your direct conversational style.
> > > > > > >
> > > > > > > Thank you for your detailed responses.
> > > > > > >
> > > > > > > > 1. Zero-gradient
> > > > > > > Your explanation makes sense.
> > > > > > > In the case where GRPO cannot sample even a single correct answer, Anchor+GRPO essentially degenerates into SFT with extra negative gradients from other rollouts. Given this dynamic, I am still trying to understand why SFT+GRPO underperforms.
> > > > > > >
> > > > > > > However, I acknowledge that the zero-gradient problem of GRPO needs to be tackled.
> > > > > > >
> > > > > > > One thing I would like to note is that the zero-gradient issues might not have an impact on other policy gradient optimization without group relative advantages, e.g., PPO, ReMax...
> > > > > > > This needs to be discussed.
> > > > > > >
> > > > > > > > 2. Forcing the ground truth trajectory.
> > > > > > >
> > > > > > > I may have misinterpreted the method earlier; the technique makes sense to me now. However, I am curious about the design choice: why introduce an SFT-like token-level gradient instead of simply using the ground truth as an extra rollout with a positive advantage?
> > > > > > >
> > > > > > > Responses 3, 4, and 5 are clear to me.
> > > > > > >
> > > > > > >
> > > > > > > I have raised my score to 6 for now and look forward to the discussion with other reviewers.

---

> > > > > > > > ### Author Response · Authors · 2025-11-27
> > > > > > > >
> > > > > > > > Thank you very much for raising your score and the follow-up!
> > > > > > > >
> > > > > > > > > 1. Given this dynamic, I am still trying to understand why SFT+GRPO underperforms.
> > > > > > > >
> > > > > > > > Even when we train with SFT alone until convergence, performance remains extremely poor, such as on answerable instances in GRAPHLA. This indicates that SFT by itself does not meaningfully help the model acquire the required multi-step deductive reasoning skills. As a result, using SFT as an initialization for GRPO offers only a slight improvement but remains fundamentally similar to starting GRPO from scratch, leading to the underperformance observed.
> > > > > > > >
> > > > > > > > > 2. One thing I would like to note is that the zero-gradient issues might not have an impact on other policy gradient optimization without group relative advantages, e.g., PPO, ReMax... This needs to be discussed.
> > > > > > > >
> > > > > > > > Thank you for this feedback! We have edited our paper to clarify that this issue only affects GRPO or relevant methods in Line 755.
> > > > > > > >
> > > > > > > > > 3. I am curious about the design choice: why introduce an SFT-like token-level gradient instead of simply using the ground truth as an extra rollout with a positive advantage?
> > > > > > > >
> > > > > > > > Thank you for raising this point. This appears to be a misunderstanding. Our method does exactly what you describe: ANCHOR treats the ground-truth trajectory as an additional rollout with a positive advantage. We do not use an SFT loss.
> > > > > > > >
> > > > > > > > The distinction is that: If one injects the ground truth using SFT loss, the update would be $\nabla \log p$ scaled by a constant, independent of policy ratios. In contrast, treating the ground truth as an extra rollout, as ANCHOR does, yields an update of the form $A_{\text{clip}}^* = A^⋆ \cdot \pi_{\theta} / \pi_{\text{old}}$, which is exactly the GRPO-style policy-gradient update (with clipping), and differs fundamentally from pure SFT. We also clarify this in Line 349-350.
> > > > > > > >
> > > > > > > > Please feel free to reach out with any additional questions. We would be very happy to discuss further!

---

> > > > > > > > > ### Comment · Reviewer_n28i · 2025-11-27
> > > > > > > > >
> > > > > > > > > Thanks for the responses.
> > > > > > > > >
> > > > > > > > > > 3.
> > > > > > > > >
> > > > > > > > > If it is purely the loss in GRPO, there shall not be a $\log$ in Eq (1) (if I recall the formula of GRPO correctly).
> > > > > > > > >
> > > > > > > > > But anyway, I think it is pretty minor.

---

> > > > > > > > > > ### Author Response · Authors · 2025-11-28
> > > > > > > > > >
> > > > > > > > > > Thank you for the question! Equation (1) does include a log term. Please also see Eq 12 in [1], and Eq 20 in [2] for your reference.
> > > > > > > > > >
> > > > > > > > > > [1] Group Sequence Policy Optimization
> > > > > > > > > >
> > > > > > > > > > [2] DeepSeekMath: Pushing the Limits of Mathematical Reasoning in Open Language Models

---

> > > > > > > > > > > ### Comment · Reviewer_n28i · 2025-11-28
> > > > > > > > > > >
> > > > > > > > > > > I see. It is from the Identity Trick.
> > > > > > > > > > >
> > > > > > > > > > > Thanks for pointing that out.
> > > > > > > > > > >
> > > > > > > > > > > I have no question on it now.

---

### Official Review · Reviewer_buC7 · 2025-11-01

**Soundness:** 2
**Presentation:** 3
**Contribution:** 2
**Rating:** 2
**Confidence:** 4

**Summary:**

This paper proposes ANCHOR (Augmented with Necessary Correct and HOnest Reasoning), a method that injects ground-truth reasoning trajectories into GRPO rollouts to stabilize reinforcement learning training for mathematical reasoning tasks. The authors introduce GRAPHLA, a dataset of multi-step linear algebra reasoning problems with answerable and unanswerable instances, and demonstrate that ANCHOR outperforms standard SFT and GRPO baselines.

**Strengths:**

1. Well-motivated problem: The paper clearly articulates the instability issues in GRPO when negative rewards dominate, and the importance of honesty alignment in mathematical reasoning is well-established.
2. Controlled experimental setup: The GRAPHLA dataset provides a clean testbed for studying deductive reasoning with formally verifiable answerability, avoiding confounds from factual knowledge.
3. Clear presentation: The paper is generally well-written with clear motivation, methodology description, and experimental results.

**Weaknesses:**

1. Critical Issue: Lack of Originality and Missing Related Work. The most significant problem with this submission is that the core contribution—injecting ground-truth/reference trajectories into GRPO rollouts to stabilize training—has been previously proposed and thoroughly investigated. Most notably: LUFFY (Learning to Reason Under Off-Policy Guidance) [Yan et al., 2025, arXiv:2504.14945] proposes essentially the same approach. Also, there is no comparison with related works in experiments.
2. Limited Technical Novelty. This paper propose a straightforward approach: deterministically injecting one ground-truth trajectory per group is a natural and obvious extension of GRPO—much simpler than LUFFY's policy shaping mechanism.
3. Limited scope: Evaluation is restricted to a single task domain (linear algebra) and relatively small models (≤3B parameters)

**Questions:**

See Limitation part.

---

> ### Author Response · Authors · 2025-11-18
>
> Thank you for your feedback. In addition to our general response, we address your specific concerns below.
>
> > 1. Lack of originality, novelty, and missing related work
>
> We agree that LUFFY shares high-level similarities with our work. However, there are important methodological differences. LUFFY leverages trajectories from a teacher model rather than from ground truth, and it introduces many additional components such as regularized importance sampling and the removal of on-policy clipping. In contrast, our approach focuses on a simpler mechanism that relies on ground truth reasoning trajectories, which leads to both conceptual clarity and practical stability.
>
> Beyond methodology, we emphasize that our contribution is not limited to the modification of GRPO. A substantial part of the paper focuses on honesty alignment in deductive reasoning, including a novel task formulation and the construction of the datasets. These components represent a significant portion of the contribution and are orthogonal to LUFFY’s scope.
>
> We have expanded our discussion of related work to provide more context. While we would ideally compare against all relevant RL-based methods, the landscape is large and most prior work does not mutually compare beyond GRPO. Their reported performance differences relative to GRPO are also small. Our claim is not that ANCHOR outperforms every existing method, but rather that it provides a simple and effective solution to limitations in both SFT and GRPO under our setting. We hope this overall framing clarifies the novelty of our contribution.
>
> > 2. Limited scope of evaluation tasks and model sizes
>
> Our experiments cover two domains: linear algebra and logical inference. We intentionally focus on domains where deductive reasoning can be evaluated without reliance on external world knowledge, as our goal is to study honest deductive reasoning rather than performance on general-purpose benchmarks such as MATH500. If there exist datasets more aligned with faithful deductive reasoning, we would welcome references.
>
> Regarding model sizes, our choices were constrained by available compute resources. Scaling to substantially larger models would come at significant computational cost without necessarily yielding proportionate scientific insight for the questions we investigate. We would be glad to extend these experiments if computational resources are available.
>
> We hope these clarifications address your concerns, and we appreciate your consideration in updating the score.

---

> > ### Author Response · Authors · 2025-11-20
> >
> > Dear reviewer, we would like to kindly ask if we have addressed your concerns. We sincerely appreciate your questions and feedback!

---

> > > ### Author Response · Authors · 2025-11-21
> > >
> > > Dear reviewer, we sincerely appreciate the time you have invested in reviewing our work. If you have any further thoughts after reading our rebuttal, we would be grateful to hear them.

---

> > > > ### Author Response · Authors · 2025-11-22
> > > >
> > > > Dear reviewer, we appreciate that you may have limited time to review our responses. Please feel free to share any thoughts or suggestions at your convenience.

---

> > > > > ### Author Response · Authors · 2025-11-25
> > > > >
> > > > > Dear reviewer, we hope you are doing well. We wanted to kindly check in to see if there is anything further we can clarify regarding our work. If you find our responses helpful, we would be grateful if you would consider updating your score. Thank you again for the time and care you have dedicated to reviewing our submission.

---

> > > > > > ### Author Response · Authors · 2025-11-27
> > > > > >
> > > > > > Dear reviewer, the author-reviewer discussion phase will end soon. We warmly invite you to consider our responses and raise your scores if our responses were helpful. We are very happy to provide further information and thank you very much for your time!

---

### Official Review · Reviewer_K8Nm · 2025-11-02

**Soundness:** 3
**Presentation:** 3
**Contribution:** 3
**Rating:** 4
**Confidence:** 3

**Summary:**

This work investigates the LLM's ability to not only solve answerable problems but also to reliably identify and abstain from answering unanswerable ones through the lens of deductive reasoning. The authors introduce two novel datasets, GRAPHLA (linear algebra-based) and GRAPHLI (logical inference-based), featuring balanced instances of answerable/unanswerable questions. They propose ANCHOR (Augmented with Necessary Correct and HOnest Reasoning), which injects ground-truth trajectories into GRPO rollouts to stabilize reinforcement learning when all sampled responses are incorrect. The authors formally show that ANCHOR's gradient effectively adds a clipped, SFT-like term to the GRPO update, unifying supervised and reinforcement learning. Empirically, they demonstrate that ANCHOR successfully stabilizes training and outperforms baselines (SFT, GRPO, SFT+GRPO, and curriculum learning) on their new datasets.

**Strengths:**

1. The diagnosis of the "gradient collapse" failure mode of GRPO in the context of honesty alignment is a novel and valuable contribution. And the ANCHOR method is simple, well-motivated.
2. The heatmaps in Figure 1 demonstrate that baseline models fail at both reasoning and abstention as complexity increases.

**Weaknesses:**

1. Both datasets are synthetic and highly structured. Real-world honesty alignment involves messier scenarios where unanswerability is not binary or deterministically verifiable. The clean graph structure may not capture the complexities of knowledge boundaries in practice.
2. No comparisons to related works on stabilizing gradients, like Melo et al, 2025 [1]
3. In many real-world scenarios, a single, verifiable ground-truth reasoning path is often unavailable, expensive to annotate. This is feasible for the synthetic datasets (GRAPHLA, GRAPHLI) created by the authors.
4. Only Qwen models are tested. Experiments on other model families (Llama, Mistral, etc.) would strengthen claims.
5. No convergence guarantees are provided. Under what conditions does ANCHOR converge?

[1] Stabilizing Policy Gradients for Sample-Efficient Reinforcement Learning in LLM Reasoning https://arxiv.org/abs/2510.00819

**Questions:**

1. The tasks are extremely difficult even for humans (e.g., solving systems with 10+ variables, tracking 15-step logical chains). Models might fail not due to dishonesty, but simply because of insufficient capacity. Can this be possible?
2. Process reward models that provide step-by-step supervision could be a strong baseline.
3. Recent work on difficulty-aware RL (GRPO-LEAD) shows that careful curriculum design can significantly improve GRPO training. The learning results from the curriculum are presented as a limitation. Will techniques like this help?

[1] GRPO-LEAD: A Difficulty-Aware Reinforcement Learning Approach for Concise Mathematical Reasoning in Language Models https://arxiv.org/abs/2504.09696

---

> ### Author Response · Authors · 2025-11-17
>
> Thank you for the feedback. In addition to the general response we posted, we address your comments below:
>
> > 1. Datasets are synthetic and highly structured. The clean graph structure may not capture the complexities of knowledge boundaries in practice.
>
> As explained in the introduction, our motivation is to isolate honest deductive reasoning capability from factual recall. Realistic datasets intertwine multiple skills: reasoning, domain knowledge, linguistic patterns, and stylistic cues, making it difficult to determine the source of model errors. They also introduce contamination risks because pretraining corpora may contain similar examples. Synthetic datasets provide the essential advantage of controllability: they allow us to define tasks where answerability is determined solely by the underlying graph structure, precisely control difficulty and reasoning depth, and cleanly separate reasoning from knowledge. Our goal is not to improve performance on compositional or realistic benchmarks, but to study one specific aspect of honesty: whether a model can recognize when the premises are insufficient for a conclusion in multi-step deductive reasoning. Applying stabilization methods like ours to realistic settings is an interesting direction for future work, but it lies beyond the focus and scope of this paper.
>
> > 2. No comparisons to related works on stabilizing gradients like https://arxiv.org/abs/2510.00819
> . Additionally, GRPO-LEAD shows that careful curriculum design can significantly improve GRPO training.
>
> The paper you referenced was published after our submission deadline, and its motivation and approach fundamentally differ from ours. It focuses on addressing general RL training stability issues such as non-stationary objectives, high-variance gradients, and aggressive update regimes, rather than the zero-variance problem that anchors our approach. For this reason, the methods are not directly comparable. We have expanded the related work section to reflect a more comprehensive discussion of relevant literature.
>
> GRPO-LEAD does improve GRPO, but it relies on a large stack of carefully tuned components: multiple reward terms, numerous hyperparameters, staged curricula, and SFT, making its improvements dependent on an engineered training pipeline rather than a simple or generalizable curriculum strategy. The complexity of the method actually reinforces our claim that curriculum learning helps only when accompanied by substantial additional machinery.
>
> > 3. In real-world scenarios, a single, verifiable ground-truth reasoning path is often unavailable.
>
> Thank you for the question. The same assumption is required for supervised fine-tuning, which also depends on ground-truth reasoning trajectories. Moreover, our method naturally extends to situations where multiple valid reasoning paths exist for a single query. In such cases, all of them can be injected into the model’s rollouts.
>
> > 4. Only Qwen models are tested.
>
> We selected Qwen models because they are currently the strongest open-source reasoning models, with performance characteristics best suited for studying multi-step deductive reasoning. Other open-source families such as Llama or Mistral have not yet demonstrated comparable reasoning capability or maturity in this area. If competitive reasoning-oriented models from other families emerge and are validated by the community, we would be eager to include them in future work.
>
> > 5. No convergence guarantees are provided.
>
> We agree that theoretical convergence guarantees are valuable in principle. In the context of RL methods for LLMs, however, deriving such guarantees is challenging because practical implementations rely on techniques such as clipping and normalization that deviate from the assumptions typically required for formal analysis. As a result, any proof would necessarily apply to a highly idealized version of the algorithm rather than the one actually used in practice, limiting its relevance. The RL papers referenced in our related work section do not provide theoretical convergence guarantees, including widely used approaches such as GRPO. For this reason, we believe such guarantees would offer limited practical value.

---

> ### Author Response · Authors · 2025-11-17
>
> > 6. The tasks are extremely difficult even for humans.
>
> We agree. However, current LLMs already surpass human capabilities in several domains, and designing challenging tasks is aligned with the broader objectives of “superalignment.”
>
> > 7. Process reward models that provide step-by-step supervision could be a strong baseline.
>
> In the early stages of our work, we extensively explored various forms of process rewards, but none were successful. We attempted several settings: (1) extracting reasoning steps from trajectories using sentence boundaries, newline characters, or explicit instructions such as requiring each step to appear within `<step></step>` tags; (2) assigning outcome rewards only to tokens within `<answer></answer>` while giving separate process rewards to tokens within `<think></think>`, or weighting outcome rewards and process rewards jointly for tokens prior to `<answer>`; and (3) generating process rewards using LLM-as-a-judge, rule-based matching for intermediate variable values, and entropy-based signals.
>
> All of these attempts failed across all three model sizes and both datasets, providing no meaningful learning signal. We found it extremely difficult to design appropriate process rewards for each step, and even when a workable signal existed, it tended to be highly task-specific (e.g., differing between linear algebra and logical inference), which is undesirable for our goal of developing a generally applicable method. For this reason, we removed the discussion of process rewards in the initial submission. In the updated draft, we have added a brief discussion of these findings in Appendix H.
>
> We hope our responses address your concerns, and we would be grateful for your consideration in updating the score.

---

> > ### Author Response · Authors · 2025-11-18
> >
> > Dear reviewer, we would like to kindly ask if we have addressed your concerns. We sincerely appreciate your questions and feedback!

---

> > > ### Author Response · Authors · 2025-11-20
> > >
> > > Dear reviewer, please let us know if you have any further questions or concerns. We will be happy to address them, and we would be grateful if you would consider updating your scores.

---

> > > > ### Author Response · Authors · 2025-11-21
> > > >
> > > > Dear reviewer, we sincerely appreciate the time you have invested in reviewing our work. If you have any further thoughts after reading our rebuttal, we would be grateful to hear them.

---

> > > > > ### Author Response · Authors · 2025-11-22
> > > > >
> > > > > Dear reviewer, we appreciate that you may have limited time to review our responses. Please feel free to share any thoughts or suggestions at your convenience.

---

> > > > > > ### Author Response · Authors · 2025-11-25
> > > > > >
> > > > > > Dear reviewer, we hope you are doing well. We wanted to kindly check in to see if there is anything further we can clarify regarding our work. If you find our responses helpful, we would be grateful if you would consider updating your score. Thank you again for the time and care you have dedicated to reviewing our submission.

---

> > > > > > > ### Comment · Reviewer_K8Nm · 2025-11-27
> > > > > > > **Response to Author's Rebuttal**
> > > > > > >
> > > > > > > I thank the authors for their response. They have addressed my concerns, and I have updated my review score accordingly.

---

> > > > > > > > ### Author Response · Authors · 2025-11-27
> > > > > > > >
> > > > > > > > Thank you for raising the score! Your feedback is very valuable for improving our paper. We truly appreciate your time and thoughtful comments.

---

### Author Response · Authors · 2025-11-17

Thank you to all reviewers for taking the time to evaluate our submission. We noticed several recurring misunderstandings and questions regarding our motivation, the scope of our work, and the nature of our contributions. To address these issues and to improve clarity, we have updated the Abstract, Introduction, and Related Work sections accordingly. We provide a consolidated explanation below.

Our work is motivated by two central goals. First, we aim to design datasets for multi-step deductive reasoning in a way that reflects honest behavior: when a conclusion is logically derivable by the premises, the model should respond accurately, and when it is not valid, the model should abstain rather than fabricate information. Second, we aim to develop a method that can combine the advantages of GRPO and SFT, stabilizing policy gradient training while preserving the benefits of relative credit assignment. It is important that our contributions be understood through these two lenses, and we have revised our manuscript to make these motivations more explicit.

1. A common question raised in the reviews is why we do not evaluate our method on real-world honesty-related datasets. Our research is specifically focused on honesty in deductive reasoning, which is proposed to isolate deductive structure from factual recall and other confounding factors. Realistic datasets tend to mix multiple skills: reasoning, domain knowledge, linguistic expectations, even stylistic cues, making it unclear what type of error a model makes when it fails. They could also introduce contamination risks because pretraining corpora may include similar examples. For this line of inquiry, synthetic datasets offer the crucial advantage of controllability: we can define tasks where the answerability is fully determined by the underlying graph structure, precisely manipulate difficulty and reasoning depth, and ensure clean separation between reasoning and knowledge. Our goal is not to improve performance on compositional realistic dataset benchmarks, but to understand and study one specific form of honesty: whether a model can recognize when premises are insufficient for a conclusion in multi-step deductive reasoning.

2. Another question concerns our definition of honesty. Honesty includes two dimensions: (1) models should be aware of the limits of their own knowledge, and (2) they should recognize when a question is answerable from the provided information rather than fabricating conclusions. Our paper focuses on the second aspect, using deductive reasoning as a testbed where answerability is objectively defined and interpretations are unambiguous.

3. Regarding originality, our contributions extend beyond a single methodological insight. First, we design controlled, multi-step deductive reasoning datasets specifically tailored for studying honest reasoning behavior. We analyze why existing approaches such as SFT, GRPO, and various patching strategies on them fail in this setting. We provide theoretical analysis showing how ANCHOR, injecting ground-truth reasoning trajectories, stabilizes policy gradient updates by mitigating gradient vanishing, and we demonstrate empirically that this approach leads to substantial improvements. To our knowledge, no prior work addresses all of these components together. The simplicity of our method is intentional, not a drawback: if a simple mechanism can resolve instability and enable models to learn long-range deductive patterns, introducing additional modules or constraints would add unnecessary complexity and overfit the datasets tested. We hope that readers will consider the full scope of our contributions rather than isolating one dimension and over-interpreting it.

4. Why don’t we evaluate larger models or additional model families? (1) Our experiments are limited by available compute resources, and extending evaluations to substantially larger models would not offer a commensurate scientific benefit considering the cost. We would be happy to run such experiments if resources are available. (2) We chose Qwen models because they are currently the strongest open-source reasoning models, with performance profiles most appropriate for studying multi-step deductive reasoning. Other open-source families such as Llama or Mistral have not demonstrated comparable reasoning ability or maturity in this domain. If competitive reasoning-oriented models emerge from other families and have been thoroughly validated in other studies, we are eager to include them in future work.

---

> ### Author Response · Authors · 2025-11-17
>
> For transparency, we summarize the primary changes in the revised manuscript:
> - The Abstract now communicates our motivation more clearly.
> - The Introduction includes a more explicit explanation of our goals and an expanded discussion relating our method to prior work.
> - The Related Work section has been moved into the main text, streamlined to remove less relevant citations, and augmented with more pertinent literature.
> - Appendix A now contains clarifications on several overloaded concepts to help readers understand our definitions and assumptions.
>
> In summary, our paper introduces the concept of honesty in deductive reasoning, constructs two controllable multi-step deductive reasoning datasets, analyzes the limitations of existing approaches such as SFT and GRPO, and proposes ANCHOR: a simple yet effective method for unifying SFT and GRPO signals by injecting ground-truth trajectories into policy rollouts. Through theoretical insights and extensive empirical evidence, we show that ANCHOR stabilizes training, mitigates gradient vanishing, and significantly improves both deductive accuracy and honest abstention.
>
> Please feel free to comment in this thread if any part of the explanation above remains unclear. We are happy to provide further clarification.

---

### Meta-Review · Area_Chair_XSLk · 2026-01-07

**Summary:**

This paper studies deductive reasoning for LLMs --- answer correctly when the conclusion is logically entailed by the provided premises, and abstain (unknown/no) otherwise --- using two controlled, synthetic multi-step benchmarks (GRAPHLA for linear algebra-style equation chains; GRAPHLI for propositional inference chains).

The main technical contribution an algorithm that deterministically injects a ground-truth trajectory into each GRPO rollout group to avoid the the ``all rollouts bad -> zero group-relative advantage -> no learning signal / collapse'' regime; the paper also provides a formal derivation showing this adds an SFT-like gradient term while retaining GRPO-style clipping and relative credit assignment. Empirically, the authors show strong gains over prompting, SFT, GRPO, and SFT+GRPO, and demonstrate that curriculum learning can work but is brittle to difficulty calibration (with ANCHOR often more robust and complementary).

Across reviews, there is broad agreement on the clear motivation (honesty via abstention under deductive entailment), clean experimental controllability (difficulty knobs), and a simple stabilization mechanism that materially improves training dynamics and end performance (particularly where GRPO collapses to ``always abstain'').

The main concerns
* generalizability of the performance to other tasks (given the synthetic, structured tasks considered in paper)
* originality vs closely related work on off-policy guidance and demonstration injection (notably LUFFY raised by one reviewer) and lack of direct experimental comparison;
* realism of assuming access to ground-truth trajectories during RL
* questions on RL hyperparameter choices and counterintuitive SFT+GRPO behavior.

The rebuttal/discussion addressed some of these issues. Among the remaining negative ratings, the primary objections are around novelty and extended empirical results.

**Reviewer Concerns:**

Concerns addressed
* many clarity issues were addressed
* the authors clarified that the "zero-variance" issue is specific to group-relative advantage methods like GRPO

Concerns partially addressed / outstanding
* ground-truth trajectory availability assumption
* multiple reviewers question how far the synthetic graph constructions generalize to real-world settings.
* One reviewer argues the core idea resembles LUFFY and related off-policy guidance approaches, and there is no head-to-head evaluation. The rebuttal distinguishes "teacher-generated trajectories" vs ground-truth trajectories. Still, without experiments against the closest baselines (or a stronger empirical separation), this concern is only partially mitigated.

**Reviewer Scores:**

even under an extended discussion, I expect the paper would land around two positive reviews and two negative reviews rather than consensus.

---

### Decision · Program_Chairs · 2026-01-26

Reject